# Recurrence of Uterine Smooth Muscle Tumor of Uncertain Malignant Potential: A Systematic Review of the Literature

**DOI:** 10.3390/cancers14092323

**Published:** 2022-05-07

**Authors:** Jacopo Di Giuseppe, Camilla Grelloni, Lucia Giuliani, Giovanni Delli Carpini, Luca Giannella, Andrea Ciavattini

**Affiliations:** Woman’s Health Sciences Department, Gynecologic Section, Polytechnic University of Marche, Via Filippo Corridoni, 16, 60123 Ancona, Italy; jacopodigiuseppe@gmail.com (J.D.G.); camygrello@googlemail.com (C.G.); lucia-giuliani@alice.it (L.G.); giovanni.dellicarpini@ospedaliriuniti.marche.it (G.D.C.); lucazeta1976@libero.it (L.G.)

**Keywords:** uterine STUMP, fibroid uterus, smooth muscle tumor, uncertain malignant potential, leiomyoma, leiomyosarcoma

## Abstract

**Simple Summary:**

The preoperative diagnosis of the uterine smooth muscle tumor of uncertain malignant potential (STUMP) is a challenge, as it does not have specific preoperative features; diagnosis is often incidental to the postoperative specimen in patients with suspected fibroids. Electrical morcellation during laparoscopic surgery can lead to cellular diffusion in the peritoneal cavity. Therefore, in 2014, the U.S Food and Drug Administration published an alert that focused the entire world’s attention on the associated risks. Although STUMPs are thought to have low malignant potential, their capacity for recurrence or metastasis is described. Nevertheless, there are no available guidelines, and data about the risk factors for recurrence are scarce and of low quality. Thus, we felt the need for an updated literature review on this topic to provide any information about characteristics and outcomes of patients with STUMP over a period of 60 years, focusing on the risk factors for recurrence.

**Abstract:**

**Background**: This study aimed to systematically review the existing literature on uterine smooth muscle tumor of uncertain malignant potential (STUMP) to provide information about characteristics and outcomes of patients and the risk factors for recurrence over a period of 60 years (1960–2021). **Methods:** According to PRISMA guidelines, we searched for "uterine smooth muscle tumor of uncertain malignant potential" in PubMed (all fields) and Scopus (Title/Abstract/Keywords) databases (accessed on 1 January 2022). Relevant articles were obtained in full-text format and screened for additional references. The only filter used was the English language. Studies including full case description of patients with histopathological diagnosis of STUMP in accordance with Stanford criteria were included. **Results:** Thirty-four studies, including 189 cases, were included. The median age was 43 years, and in 21.5% of cases there was a recurrence of the disease. Bivariate analysis showed a significant association between use of morcellation without bag and risk of recurrence (*p* = 0.001). Unprotected morcellation during demolitive or conservative surgery was independently associated with a higher risk of disease recurrence with a relative risk of 2.94 (*p* < 0.001). A significant progressive decrease in the recurrence rate was observed over time (r = −0.671, *p* = 0.008). The percentage of patients who underwent surgery followed by in-bag protected morcellation significantly increased after the publication of the U.S. Food and Drug Administration alert about the risk linked to this procedure (*p* = 0.01). **Conclusions:** Unprotected morcellation of the lesion is associated with the relapse of the disease. However, this clinical condition showed a drastic decrease over time. This could likely be due to the increased awareness by surgeons of the importance of customizing surgical treatment.

## 1. Introduction

Uterine fibroids are the most common benign tumors of the urogenital system in the female population [1]. Sometimes, they are associated with a complete absence of symptoms, thus allowing a wait-and-see behavior. In other patients, the onset of signs and symptoms (hypermenorrhea, menometrorrhagia, pelvic pain, or infertility) can require medical treatment or, in most cases, a surgical approach such as myomectomy or hysterectomy [2]. The choice of the surgical technique depends primarily on the type of the lesion (s), the characteristics of the patient, and the experience of the surgeon. The laparoscopic approach, when performed by expert operators, is associated with demonstrated advantages [3,4], but electrical morcellation during laparoscopic surgery can lead to cellular micro diffusion in the peritoneal cavity, with the possibility of implantation in ectopic sites. These events are reported especially when the primary lesion is an unexpected malignant tumor. The postoperative histological malignant diagnosis rate ranges from 0.08 to 0.13% [5,6,7,8,9,10,11,12,13], thus worsening the prognosis. Therefore, in 2014, the U.S Food and Drug Administration (FDA) published an alert that focused the entire world’s attention on the risks associated with morcellation [14,15]. 

Although showing good accuracy, diagnostic imaging techniques do not always allow a preoperative diagnosis of malignancy [16,17,18,19,20,21]. Likewise, the preoperative diagnosis of the smooth muscle tumor of uncertain malignant potential (STUMP) is a challenge, as it does not have specific radiologic features. Under the definition of the World Health Organization (WHO) classification [22], a STUMP can be defined as a uterine smooth muscle tumor that cannot be diagnosed unequivocally as benign or malignant. Although STUMPs are thought to have low malignant potential, their capacity for recurrence or metastasis is described. For this reason, the diagnosis of STUMPs is often incidental to the postoperative specimen in patients with suspected fibroids [23]. 

To date, there are no available guidelines, and data regarding the risk factors for recurrence are scarce and of low quality. 

The purpose of this study was to systematically review the existing literature on STUMPs to provide information about characteristics and outcomes of patients, and the risk factors of recurrence, over a period of 60 years (1960–2021).

## 2. Materials and Methods

The Preferred Reporting Items for Systematic Reviews and Meta-Analyses (PRISMA) guidelines were followed to systematically review the literature by searching the PubMed and Scopus databases [24]. The review protocol was registered on Research Registry (https://www.researchregistry.com, accessed on 15 January 2022) (reviewregistry1353). The present study assessed the following (Population, Intervention, Comparison, Outcomes (PICOS) questions: Population: women diagnosed with STUMP.Intervention: surgical intervention (myomectomy or hysterectomy) performed for uterine mass or related symptoms.Comparison: patients with recurrence of disease and without recurrence.Outcomes: (1) descriptive analysis of patient characteristics, also evaluating the features associated with the risk of relapse, (2) distribution of the recurrence rate over time, (3) follow-up outcomes when available.

Study design: a systematic review of retrospective observational studies (case reports, case series).

Eligibility/inclusion criteria: studies including full case description of patients with histopathological diagnosis of STUMP. The histologic criteria used for the classification of uterine smooth muscle neoplasms were in accordance with Stanford criteria, published in 1994 [25]. The first definition of STUMP was published by the WHO in 2003 [26]. 

Exclusion criteria: Review articles were excluded; research articles on case series without a full case description or in a non-English language were excluded. 

Information sources and search strategy: We searched for “uterine smooth muscle tumor of uncertain malignant potential” in PubMed (all fields) (accessed on 1 January 2022) and Scopus (Title/Abstract/Keywords) (accessed on 1 January 2022) databases. Relevant articles were obtained in full-text format and screened for additional references. The only filter used was the English language.

Study selection: Three independent reviewers (Jacopo Di Giuseppe, Camilla Grelloni, Lucia Giuliani) selected the studies using a two-step screening method. At first, the screening of titles and abstracts was performed to assess eligibility and inclusion criteria and exclude irrelevant studies. Afterward, the three reviewers evaluated full texts of included articles to (1) assess study eligibility and inclusion criteria and (2) avoid duplications of the included cases. The same authors manually searched reference lists to search for additional relevant publications. Andrea Ciavattini and Jacopo Di Giuseppe checked the data extracted. 

The objective of this systematic review was: (1) to provide any information about characteristics and outcomes of patients with STUMP diagnosis over a 60-year period (1960–2021); (2) to provide information about the risk factors for recurrence and about the distribution of the recurrence rate.

Data collection process/data items: Data collection was study-related (authors and year of study publication) and case-related: age at surgery, parity, previous surgery, history of infertility, BMI, presence of symptoms or signs, cancer antigen 125 (CA-125), ultrasound features of the tumor, largest tumor size (cm), STUMP localization, type of surgery, use of morcellation were evaluated). Moreover, histological features, such as the number of mitoses, the degree of atypia, and the presence of necrosis, were recorded. Outcomes of evaluated patients were need for second surgery, possible presence, and site of recurrence, surgical treatment, and histology of recurrence.

Statistical analysis: The collected data were reported as continuous or categorical variables. Continuous variables were tested for normal or not-normal distribution using the D’Agostino Pearson test. According to distribution, they were expressed as mean ± standard deviation or median and range. Categorical variables were expressed as frequency and percentage and were compared with Chi-square or Fisher’s exact test. A bivariate analysis was performed to identify factors that were significantly associated with the risk of recurrence and with the risk of malignant transformation of recurrence. The relative risk was used to evaluate the association between the use of morcellation and the risk of recurrence. Correlation analysis was used to determine whether the values of two variables were associated (if they had normal distribution). When the distribution of variables was not normal, the relationship between the variables was determined using Rank correlation (the Spearman’s coefficient). A *p*-value of less than 0.05 was considered statistically significant.

MedCalc^®^ Statistical Software version 20 (MedCalc Software Ltd., Ostend, Belgium; https://www.medcalc.org; 2021 (accessed on 15 January 2022) was used. 

## 3. Results

### 3.1. Literature Review Details

We retrieved 1584 articles on PubMed and 266 papers on Scopus databases (accessed on 1 January 2022); then, we removed 260 duplicate records. Based on title and abstract, 1515 records were excluded. Then, the full text of 75 papers was evaluated for eligibility. Based on inclusion and exclusion criteria, 41 articles were further removed. Finally, 34 studies published in the period 2004–2021 were assessed for qualitative synthesis, including 189 cases (Figure 1). All patients met the inclusion criteria, and clinical, histological, and treatment data were collected. All studies and patients are detailed in Appendix A [7,27,28,29,30,31,32,33,34,35,36,37,38,39,40,41,42,43,44,45,46,47,48,49,50,51,52,53,54,55,56,57,58,59].

### 3.2. Patient Characteristics

Patient characteristics are described in Table 1. The median age was 43 years (range: 18–75). Most of the women were parous (69%), while 6.8% of patients suffered from infertility. The risk factors for STUMP are unknown: 41.9% of women had BMI > 30 kg/m^2^, and only 12.5% of cases had already undergone previous gynecological surgery (myomectomy). CA 125 levels were less than 35 U/L in 87.9% of cases. 

The symptoms and signs mostly described were the same as those caused by fibroids: abnormal uterine bleeding (AUB), menorrhagia, dysmenorrhea, abdominal/pelvic pain, compression of adjacent organs, anemia, and pelvic mass. Of the patients, 82.5% presented at least a large uterine mass ≥ 5.0 cm of diameter. Many patients with STUMP (44.3%) underwent a surgical procedure after an incidental diagnosis of uterine mass during routine gynecological check-ups. 

### 3.3. STUMP Lesion Characteristics

Table 2 shows the characteristics of STUMP lesions. In all cases, the diagnosis of STUMP was made with the histological examination, following the criteria of the WHO classification [22]. For each lesion, the main microscopic characteristics were analyzed, including the number of mitoses, the grade of cell atypia, and the presence or absence of tumor necrosis. The median diameter of the STUMP was 8.0 cm (range: 0.7–39.0 cm). Jang et al. [30] described a case of a 41-year-old patient who came to their attention for abdominal pain, asthenia, and dysmenorrhea diagnosed with a STUMP with a diameter of 39.0 cm. Radiological investigations did not reveal any suspicion of malignancy, and the woman underwent a laparotomy hysterectomy. The patient was followed up for 18 months and suffered no postoperative complications or recurrence. In most cases of our study, preoperative radiological investigations revealed a uterine mass without suspicion of malignancy, and the diagnosis of certainty was made only after histological examination. 

### 3.4. Treatment

Table 3 shows the surgical treatment of patients included. The primary surgery was myomectomy for 79 patients (42%), total hysterectomy with bilateral adnexectomy (TH + BSO) for 60 patients (32%), and TH for 50 patients (26%). At least one previous pregnancy was reported in 89.3% of patients who underwent hysterectomy and only in 45% of patients who underwent myomectomy.

Among the cases subjected to myomectomy, in 16 cases (20.3%) morcellation was performed, in 34 cases (43%) it was not performed or was made in a bag, and in 29 cases (36.7%) this technical information was not provided. In 7 cases, morcellation was performed during hysterectomy: 6 cases during laparoscopic TH, and 1 case during laparoscopic TH + BSO.

In addition, once STUMP was diagnosed, 21.5% of myomectomy patients (17 cases) underwent a second surgery: TH + BSO in 9 cases (53%) and TH in 8 cases (47%). In 62 cases (78.5%), no secondary surgery was performed.

### 3.5. Recurrence

In 37 cases (19.5%), there was a recurrence of the disease (Table 4). A local recurrence was recorded in 23 cases (62.2%): 14 cases of pelvis recurrence and 9 cases of uterine recurrence. In the remaining 14 (37.8%) cases, the recurrence occurred in distant organs (3 lung recurrences, 3 abdominal recurrences, and 2 recurrences in other organs). In 6 cases, data about the site of recurrence were missing. 

At the histological examination of the recurrence, the diagnosis of STUMP was reconfirmed in most of the patients (67.6%), while in the remaining 12 cases (32.4%) a malignant evolution of the disease was noticed. Bivariate analysis comparing women with and without a malignant transformation of recurrence did not show a significant association between STUMP diameter, histological features, or demographic characteristics and risk of malignancy (data not shown). Eleven patients received the diagnosis of leiomyosarcoma, while 1 patient developed a liposarcoma. In this case, described by Karataşlı et al. in 2019, a 54-year-old obese woman underwent a laparotomy with TAH + BSO for AUB [34]. The histological examination resulted in a STUMP with intermediate microscopic characteristics (low degree of mitosis, moderate cellular atypia, and present necrosis). Thirty-three months after surgery, she developed pelvic pain and a pelvic mass (liposarcoma) associated with lung metastases was found. She died from the disease 62 months after the initial diagnosis.

The median follow-up of patients with recurrence was 40 months (range: 2–288). During follow-up, 4/20 patients (20%) with available data died from leiomyosarcoma (3 cases) or STUMP (1 case).

Bivariate analysis comparing women with and without recurrence (Table 5) showed a significant association between use of morcellation without bag and risk of recurrence (*p* = 0.001). 

Unprotected morcellation was independently associated with a higher risk of disease recurrence after demolitive or conservative surgery with a relative risk (RR) of 2.94 (95% confidence interval [CI] 1.6172 to 5.3785, *p* < 0.001). About 57% and 44% of patients who respectively underwent total hysterectomy with unprotected morcellation or myomectomy with unprotected morcellation experienced a recurrence. Unprotected morcellation was independently associated with a higher risk of disease recurrence in patients who underwent total hysterectomy with RR of 4.89 (95% confidence interval [CI] 2.0117 to 11.8814, *p* < 0.001) or myomectomy with RR of 1.65 (95% confidence interval [CI] 0.7508 to 3.6382, *p* = 0.21).

According to the years of case publication, a significant progressive decrease in the recurrence rate was observed over time (r = −0.671, *p* = 0.008; C.I.: −0.886 to −0.218) (Figure 2). Moreover, the rate of recurrence showed a significant decrease after 2014, the year of publication of the alert by the U.S. Food and Drug Administration (FDA) about the risk linked to the use of morcellation (*p* = 0.03; C.I.: 0.8053% to 33.1464%) (Table 6). 

A progressive decreasing rate of primary demolitive surgery (total hysterectomy with or without BSO) was also observed over the years (r = −0.662, *p* = 0.009; C.I.: −0.8826 to −0.12027), while the rate of conservative treatment (myomectomy) remained substantially unchanged (r = 0.261, *p* = 0.388; C.I.: −0.338 to 0.710). By analyzing the distribution of primary demolitive or conservative surgery before and after 2014, no significant rate differences were observed: 28.4% vs. 17.6%, for hysterectomy (*p* = 0.19), and 32.4% vs. 43.9%, for myomectomy (*p* = 0.21). 

The percentage of patients who underwent surgery followed by in-bag protected morcellation or with no morcellation, significantly increased after 2014: 63.0% vs. 41.0% (*p* = 0.01) (CI: 3.6367% to 38.4437%).

### 3.6. Missing Data

In total, 189 patients were included in this systematic review. Data on parity were available in 87 patients (Table 1). History of infertility and previous cesarean section or previous pelvic surgery were available for 59, 25, and 32 cases, respectively (Table 1). BMI was specified only in 31 women, and CA 125 level was reported in 33 patients (Table 1). Symptoms and signs were available for 151 and 44 cases, respectively (Table 1). The duration of follow-up of the entire group was determined for 148 cases (Table 1). STUMP localization was specified for 172 cases (Table 2). The duration of follow-up of recurrent cases was determined for 34 cases, while the outcome was recorded for 20 cases.

## 4. Discussion

In the current systematic review, we performed a descriptive analysis of patient characteristics, and we evaluated risk factors for STUMP recurrence. Moreover, when available, we analyzed the follow-up outcomes of patients, and we observed the distribution of the recurrence rate. 

Patients with STUMPs had symptoms like those that cause fibroids: abnormal uterine bleeding, menorrhagia, dysmenorrhea, abdominal pain, asthenia, or a combination of these. In about 45% of our cases, they were found incidentally during a routine gynecological check-up, and in about 80% of cases, patients showed a large uterine mass (≥5.0 cm of diameter) [60].

For these reasons, and given the absence of specific imaging, the diagnosis of STUMP is often post-operative [23]. Magnetic resonance (RMN) and pelvic ultrasounds are not reliable methods to pre-operatively differentiate benign from malignant tumors [29] or characterize smooth muscle uterine tumors [61]. The usefulness of ^18^fluorodeoxyglucose [FDG] positron emission tomography (PET) and computed tomography (CT) scans is debated, and few studies have been published [45,62,63,64].

The median age at diagnosis was 43 years, in agreement with data already published in the literature (mean/median age of patients: 41/48 years) [61]. The mean diameter of lesions was 8.0 cm, and most STUMPs showed intramural localization, a low or moderate number of mitoses, mild or moderate atypia, and absent necrosis, according to Stanford criteria [25]. Guntupalli et al. [65] proposed their institutional criteria, identifying five pathological categories based on prior work published by Bell et al. [25] and their institutional unpublished criteria. However, the usefulness of this more recent classification seems limited because the authors have not identified common histological features in patients with or without recurrence.

In our review, about 60% of the patients underwent primary demolition surgery, while about 40% had a primary conservative surgery (myomectomy). Among women who initially received conservative surgery, about a quarter chose to have a subsequent hysterectomy after being diagnosed with STUMP. Thus, in total, approximately 70% of the patients underwent demolition surgery, in agreement with recent literature [34]. 

Treatment planning for STUMP suffers from the inability to make an accurate preoperative diagnosis, and therapeutic choice after STUMP detection is based on patient counseling and the desire to preserve fertility, as there are no specific guidelines. The percentage of patients who had a hysterectomy was conditioned by their personal history: three-quarters of our patients reported at least one previous pregnancy, and probably they were no longer desirous of fertility: about 90% of patients undergoing hysterectomy had at least one pregnancy, while only 45% of patients undergoing myomectomy reported at least one pregnancy. 

Interestingly, 20% of patients who underwent a conservative approach were subjected to unprotected morcellation. Sometimes, this unprotected procedure was performed during the hysterectomy, too. As far as we know, this is the first review that reports data regarding the use of morcellation in such a large series of cases. Although these data are very important, unfortunately they have not been described in all published studies.

Relapse of disease occurred in about 20% of our cases. This percentage agrees with the review by Gadducci [61], who estimated the STUMP recurrence rate varying in the literature from 0 to 36%. It is important to underline that the rate of recurrence is very heterogeneous among studies because of the difficult histological diagnosis, the limited number of patients, and the different follow-up periods.

Our patients with recurrence developed local recurrence in about 60% of cases, and distant recurrence in the other cases, especially in the lung and abdomen. These data confirm that the site of relapse is often unpredictable, as published by Rizzo et al.: 33% of patients recurred locally, 33% in the lungs, and 15% in the bone [66].

Among the sites of local recurrence, particular attention should be paid to the uterus, which represents one-third of local recurrences. During follow-up of patients who have had a conservative surgical approach, it is essential to exclude the appearance of new uterine masses or the growth of lesions already existent, with pelvic examination and transvaginal ultrasound. However, less frequently, much attention must be paid to the evaluation of emerging symptoms that can lead to the diagnosis of distant relapses.

Moreover, the histological type of recurrence is variable (leiomyosarcoma in about 32% of cases), influencing the subsequent therapeutic choices, ranging from simple surgical excision to oncological treatments. Progression with malignant transformation presents with variable histology (leiomyosarcoma or liposarcoma), without the possibility of identifying risk factors for degeneration. Shapiro et al. described a case of a 46-year-old woman who was diagnosed with a “leiomyoma of uncertain malignant potential” after hysterectomy [59]. Fifty-one months later, she had a humeral fracture and during follow-up, a 10.5-cm lytic lesion adhering to her right humerus was discovered and turned out to be a leiomyosarcoma of uterine origin. Twelve months later, a CT scan showed multiple lung nodules compatible with metastasis. 

Our bivariate analysis of risk factors for recurrence showed that unprotected morcellation of the lesion excision is associated with relapse of the disease. Interestingly, about half of patients who underwent hysterectomy and myomectomy with unprotected morcellation experienced a local or distant recurrence. This is in accordance with previously published studies showing a correlation between unprotected morcellation and the increased risk of intra-abdominal recurrence in patients affected by unexpected leiomyosarcoma or STUMP [67,68,69,70,71]. Bogani et al. reported that patients with morcellated occult leiomyosarcoma had a higher risk of general and abdominal recurrence, as well as a higher mortality rate [67]. 

Most of the studies on the risk related to morcellation were published after the FDA alert, which may have influenced the therapeutic choices following its publication in 2014. The analysis of the disease recurrence trend in our study shows a significant decrease in the recurrence rate between 2004 and 2021. Furthermore, the recurrence rate significantly decreased (halved) after 2014, considering the years of case publication, suggesting a positive effect of the FDA alert on the recurrence risk.

Surprisingly, the patient’s disease-free survival rate is conditioned not by the surgical approach (laparotomic vs. laparoscopic or conservative vs. demolitive approach), but only by unprotected morcellation that is just related to minimally invasive surgery. Furthermore, even the histological characteristics do not seem to determine the increased risk of recurrence. These data do not seem to confirm what was recently published by the fifth edition of the WHO Classification of Female Genital Tumors, where a recurrence risk stratification related to necrosis, atypia, and mitosis was proposed [72].

The main strengths of this study were its systematic review of the literature, collecting a large number of case series, firstly showing that the recurrence rate of STUMP is simply related to unprotected morcellation but decreased after the 2014 FDA alert, suggesting the importance of adhering to the clinical recommendations. The limitations of this study are the missing data on recurrence features in many cases, the retrospective design of the study, and the absence of guidelines for STUMP management, also considering the rarity of STUMP.

Considering the data collected in this systematic review, the surgical treatment of STUMP should always be well planned, although this plan might be developed after a first surgery. The factors to be taken into consideration are the patient’s age, her desire for fertility, and the histological characteristics of the lesion, although future research needs to examine more closely the histological findings associated with relapse to better plan personalized surgical options and follow-up.

Preoperative features are not indicative of malignancy, and unprotected morcellation should always be avoided. Currently, this procedure is the only risk factor found, while no impact is related to the choice between a demolitive or a conservative approach. In the future, it will be essential to define ultrasound characteristics that preoperatively identify uterine masses suspected of malignancy. A recent study by Russo et al. showed that the intralesional and circumferential vascularity of the lesion, cystic areas, and the size of the mass when combined with the age of the patient can be good ultrasound parameters to distinguish benign myomas from malignant lesions [73]. Moreover, Ho et al. in 2018 proposed the “hollow ball” sign on the FDG PET, representing a zone of coagulative necrosis typical of malignancy, as a useful method to differentiate myoma from LMS/STUMP lesions [64].

## 5. Conclusions

In conclusion, the undeniable decreasing trend in the risk of recurrence underlines the increased awareness by surgeons of the importance of customizing surgical treatment. 

The systematic analysis of the risk factors associated with the relapse of STUMP is a clinical challenge to be explored further.

## Figures and Tables

**Figure 1 cancers-14-02323-f001:**
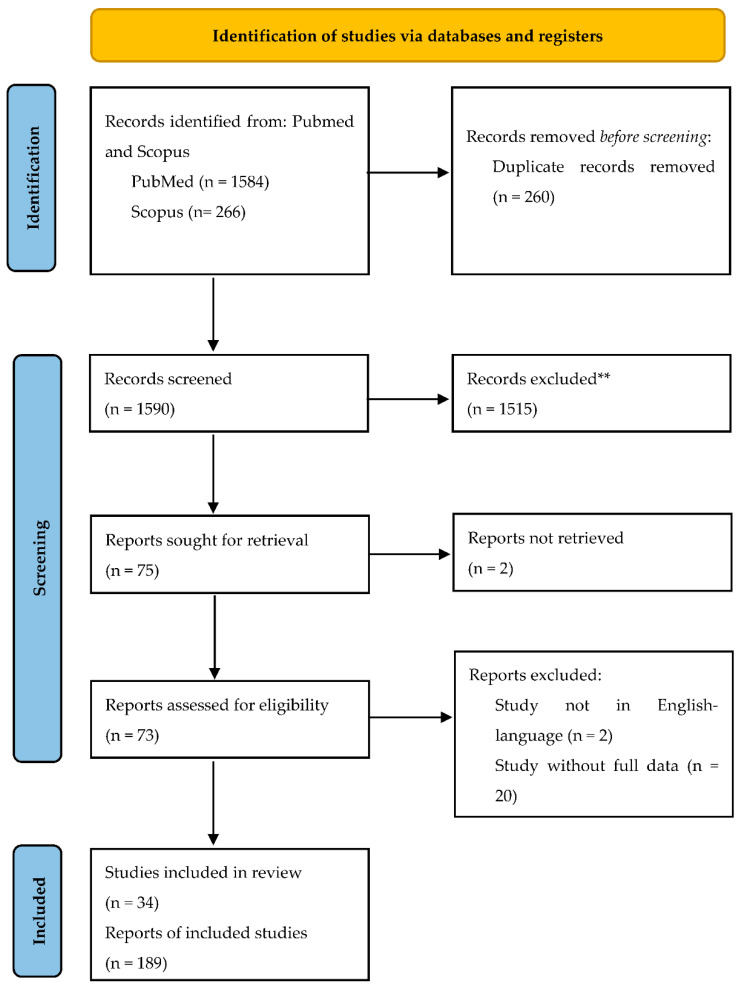
Literature review flow-chart.

**Figure 2 cancers-14-02323-f002:**
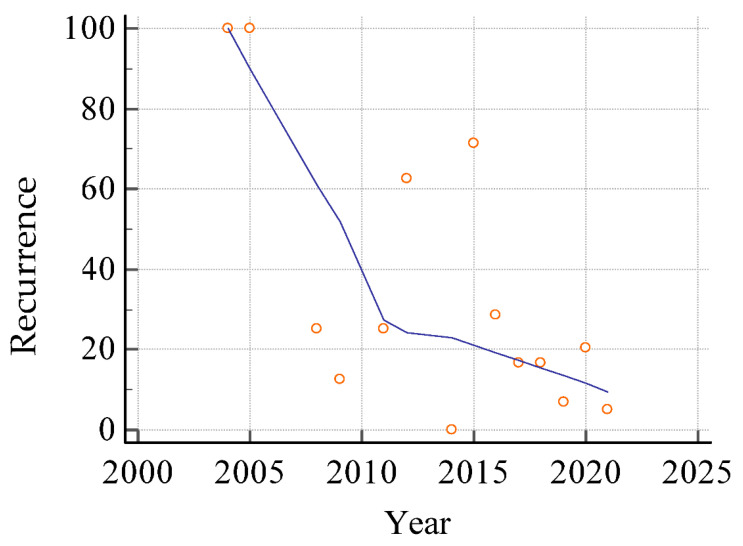
Scatter diagrams showing the trend over time of rate of recurrence (*p* = 0.008).

**Table 1 cancers-14-02323-t001:** Patient characteristics.

Characteristics	Sample Size (189) Available Data (%)
Age (years) (median, range)	43 (18–75)
Nulligravid	27/87 (31.0%)
Previous gynecological surgery (myomectomies)	4/32 (12.5%)
Previous cesarean section	4/25 (16.0%)
History of infertility	4/59 (6.8%)
BMI > 30 kg/m^2^	13/31 (41.9%)
**Initial symptoms (multiple choice)**	
AUB	41/151 (27.1%)
Menorrhagia	28/151 (18.5%)
Dysmenorrhea	4/151(2.6%)
Abdominal pain	13/151 (8.6%)
Incidentally detected	67/151 (44.4%)
Asthenia	1/151 (0.7%)
Compression of adjacent organs	7/151 (4.6%)
**Initial signs (multiple choice)**	
Anemia	9/44 (20.4%)
Pelvic mass	38/44 (86.4%)
CA125 levels < 35 U/L	29/33 (87.9%)
Follow-up duration (months) (median, range)	48 (1–288)

BMI: body mass index; AUB: abnormal uterine bleeding.

**Table 2 cancers-14-02323-t002:** STUMP lesion characteristics.

Characteristics	Sample Size (189) Available Data (%)
Diameter (cm) (median, range)	8.0 (0.7–39.0)
**STUMP localization**	
Intramural	65/172 (37.8%)
Subserosal	18/172 (10.5%)
Submucosal	12/172 (6.9%)
Intramural–Subserosal	2/172 (1.2%)
Unknown	75/172 (43.6%)
**Mitosis**	
0–4	81/189 (42.9%)
5–9	68/189 (36.0%)
≥10	15/189 (7.9%)
Unknown	25/189 (13.2%)
**Atypia**	
Severe	38/189 (20.1%)
Mild	53/189 (28.0%)
Moderate	54/189 (28.6%)
None	18/189 (9.5%)
Unknown	26/189 (13.8%)
**Necrosis**	
Absent	89/189 (47.1%)
Present	73/189 (38.6%)
Unknown	27/189 (14.3%)

**Table 3 cancers-14-02323-t003:** Treatment.

Characteristics	Sample Size (189) Available Data (%)
**Primary surgery**	**189 cases**
**Myomectomy**	**79/189 (41.8%)**
Laparoscopic myomectomy	27/79 (34.2%)
Laparotomic myomectomy	28/79 (35.4%)
Hysteroscopic myomectomy	1/79 (1.3%)
Unknown	23/79 (29.1%)
**TH**	**50/189 (26.5%)**
Laparoscopic TH	12/50 (24%)
Laparotomic TH	35/50 (70%)
Unknown	3/50 (6%)
**TH + BSO**	**60/189 (31.7%)**
Laparoscopic TH + BSO	7/60 (11.7%)
Laparotomic TH + BSO	40/60 (66.6%)
Unknown	13/60 (21.7%)
**Secondary surgery**	**Sample size (17/79) (21.5%)**
TH	8/17 (47%)
TH + BSO	9/17 (53%)

TH: total hysterectomy; TH + BSO: total hysterectomy + bilateral adnexectomy.

**Table 4 cancers-14-02323-t004:** Recurrence of disease (37 cases).

Characteristics	Sample Size (37) Available Data (%)
**Local recurrence**	**23 cases (62.2%)**
Uterus	9/23 (39.1%)
Pelvis	14/23 (60.9%)
**Distant recurrence**	**14 cases (37.8%)**
Lung	3/14 (21.4%)
Abdomen	3/14 (21.4%)
Other organs	2/14 (14.3%)
Missing data	6/14 (42.9%)
**Surgical treatment of recurrence**	**37 cases**
Debulking + chemotherapy	2/37 (5.4%)
Hysterectomy	7/37 (18.9%)
Mass excision	17/37 (45.9%)
Myomectomy	2/37 (5.4%)
TH + BSO	4/37 (10.8%)
Unknown	5/37 (13.6%)

TH + BSO: total hysterectomy + bilateral adnexectomy.

**Table 5 cancers-14-02323-t005:** Bivariate analysis comparing women with and without recurrence.

Characteristics	Recurrence (37 Cases) Available Data (%)	No Recurrence (152 Cases) Available Data (%)	*p*	C.I.
Age (median, years)	46.0	43.0	0.35	-
Mean tumor size (cm)	9.0	8.0	0.63	-
Previous cesarean section	-/6	4/19 (21.0%)	0.16	-
**Primary surgery**	**37 cases**	**152 cases**		
Myomectomy	18/37 (48.7%)	61/152 (40.1%)	0.34	-
TH	10/37 (27.0%)	40/152 (26.3%)	0.93	-
TH + BSO	9/37 (24.3%)	51/152 (33.6%)	0.27	-
Laparoscopy	13/31 (41.9%)	33/119 (27.7%)	0.12	-
Laparotomy	17/31 (54.9%)	86/119 (72.3%)	0.06	-
Hysteroscopy	1/31 (3.2%)	-	-	-
Morcellation	11/29 (37.9%)	12/105 (11.4%)	0.001	9.523–45.184
Mitosis ≥ 10	2/26 (7.7%)	13/138 (9.4%)	0.78	-
Atypia (severe)	7/25 (28%)	31/138 (22.5%)	0.55	-
Necrosis (present)	16/25 (64%)	80/137 (58.4%)	0.60	-
**Secondary surgery**	**3 /37 (8.1%)**	**14/147 (9.5%)**	**0.79**	**-**
Myomectomy	-	-	-	-
TH + BSO	3/37 (8.1%)	6/147 (4.1%)	0.31	-
TH	0/32	8/147 (5.4%)	0.18	-

TH: total hysterectomy; TH + BSO: total hysterectomy + bilateral adnexectomy.

**Table 6 cancers-14-02323-t006:** Rate of recurrence according to the years of case publication.

Outcomes	2004–2013 (n: 34) (%)	2014–2021 (n: 155) (%)	*p*	C.I.
Recurrence	11 (32.3)	26 (16.7)	0.03	0.8053% to 33.1464%

## Data Availability

The data supporting the findings of this study are available within the article and its Appendix A.

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
