# Peer review of "Recurrence of Uterine Smooth Muscle Tumor of Uncertain Malignant Potential: A Systematic Review of the Literature"

_cancers, 2022, doi:10.3390/cancers14092323_

Round 1

Reviewer 1 Report

In this article, the authors conduct an extensive research through last 60 years in order to provide some light on STUMP behaviour. Firstly, I would like to congratulate the reviewers on such a big amount of work. 

Abstract and introduction seem fine to me, so I won't suggest any changes here. 

Although authors followed the Equator initiative guidelines on how to perform a systematic review (so they followed PRISMA guidelines), there is one aspect in the PICO clinical question I would orientate in a different way. 

Usually, the intervention aspect of the question refers to "which intervention did the patient undergo", and not really all the possible variables that the researchers looked for. As the intervention performed was "surgery performed", I would simply go with that instead of the paragraph reflected in lines 82-86. 

The rest of the results section, as well as discussion seem fine. 

Author Response

We thank you very much for your careful reading of our manuscript and your constructive suggestion. We modified the PICO clinical questions by reporting the correct intervention, as suggested (lines 82-83).

Reviewer 2 Report

A systematic review of the literature referred to Recurrence of Uterine Smooth Muscle Tumor of Uncertain Malignant Potential” is without doubt an important article for gynecologists who treat uterine fibroids. There are no reliable diagnostic procedures which can differentiate benign from atypical and malignant tumors. This is why the surgical protocols must be strictly followed. This review reminds all of us that smooth muscle tumors of uncertain biological behavior may become complicated and sometimes even life-threatening disease.

The article is meticulously prepared, the literature systematically reviewed and the message clearly delivered. I have only two minor comments:

  1. There is an expression which is related to the morcellation in two lines of the text :

at Page 7 lines 224, 225

“In 7 cases, morcellation was performed after hysterectomy: 6 cases after laparoscopic TH, and 1 case after laparoscopic  TH+BSO.”

and Page 13 line 364. It

“Sometimes, this unprotected procedure was per- 364 formed after the hysterectomy, too.”

Comment: If hysterectomy was done,  why morcellation should be performed after. Morcellation is part of the procedure. Could wording " after" be replaced by "during"?

  1. One more comment is related to follow-up of patients.

Page 12, line 322

“The duration of follow-up of recurrent cases is determined for 34 cases, while the outcome is recorded for 19 cases.”

In the lines 271-273 it is stated that " During follow-up, 4/20 patients (20%) with available data died from leiomyosarcoma (3 cases)  or STUMP (1 case). " In the rows 322-323 the follow up is available in 34 patients, while outcome is recorded for 19 cases. These numbers do not correlate. Please, check

Author Response

We thank you very much for your careful reading of our manuscript and your constructive suggestions.

1. There is an expression which is related to the morcellation in two lines of the text :

at Page 7 lines 224, 225

“In 7 cases, morcellation was performed after hysterectomy: 6 cases after laparoscopic TH, and 1 case after laparoscopic  TH+BSO.”

and Page 13 line 364. It

“Sometimes, this unprotected procedure was per- 364 formed after the hysterectomy, too.”

Comment: If hysterectomy was done,  why morcellation should be performed after. Morcellation is part of the procedure. Could wording " after" be replaced by "during"?

Thank you for this comment, we replaced the word "after" with "during", as you suggested.

2. One more comment is related to follow-up of patients.

Page 12, line 322

“The duration of follow-up of recurrent cases is determined for 34 cases, while the outcome is recorded for 19 cases.”

In the lines 271-273 it is stated that " During follow-up, 4/20 patients (20%) with available data died from leiomyosarcoma (3 cases)  or STUMP (1 case). " In the rows 322-323 the follow up is available in 34 patients, while outcome is recorded for 19 cases. These numbers do not correlate. Please, check.

Thank you for this comment, we corrected line 344 because outcome is recorded for 20 cases.